# Medications for Managing Central Neuropathic Pain as a Result of Underlying Conditions—A Systematic Review

**DOI:** 10.3390/neurolint17050077

**Published:** 2025-05-20

**Authors:** Bjarke Kaae Houlind, Henrik Boye Jensen

**Affiliations:** Brain and Nerve Diseases, Lillebælt Hospital, 6000 Kolding, Denmark

**Keywords:** neuropathic pain, central neuropathic pain, neuralgia, systematic review

## Abstract

**Background:** This systematic review assessed the current literature regarding the analgesic treatment of central neuropathic pain (CNP) in central nervous system (CNS) conditions, such as spinal cord injuries, multiple sclerosis, post-stroke disorders, and Parkinson’s disease. The aim of this systematic review was to compare the current algorithmic treatment of CNP, which generally does not discriminate among underlying conditions, with RCTs investigating algorithm-recommended and non-algorithm-recommended drugs for differing underlying conditions. **Methods**: The PubMed and EMBASE databases were used to identify relevant randomized control trials (RCTs). MeSH terms and EmTree terms were searched as well as free text words in the title/abstract of the studies. A risk of bias tool was used to assess all included studies. **Results**: A total of 903 RCTs were identified from the initial search. Thirty-eight RCTs published between January 2002 and November 2024 fulfilled all the inclusion criteria and none of the exclusion criteria. The review investigated progressive and stable neurological diseases and conditions with associated CNP. **Conclusions**: From the majority of the included studies, the current recommended treatment algorithm seems to be effective and safe; however, the underlying condition seems to influence how the patient responds to tier-appropriate medication.

## 1. Introduction

The effective treatment of central neuropathic pain (CNP), described as pain caused by lesions or dysfunctions in the central nervous system (CNS), remains a clinical challenge [1,2]. It is a frequent complication in patients with spinal cord injuries (SCIs), multiple sclerosis (MS), and post-stroke disorders, like complex regional pain syndrome (PS-CRPS) and central post-stroke pain (CPSP), affecting 40–50% of SCI [3], 33% of MS [4], and 8% of stroke patients [5].

The mechanisms behind CNP are poorly understood. Irregular neural activity in compromised circuits, a post-lesion imbalance between facilitating and inhibiting neural pathways, and alterations in the treatment of incoming noxious and non-noxious stimuli are proposed as pathophysiologic features of CNP [6,7,8]. Clinical symptoms are presented as allodynia, hyperalgesia, burning pain, and others [3,7].

“A comprehensive algorithm for management of neuropathic pain” by Bates et al. [9] was published in 2019 combining the extensive meta-analysis by 

Finnerup et al. (2015) [10], and guidelines and recommendations of several national and international entities and endorses a general 1st–6th line of treatment in most cases of neuropathic pain.

First-line analgesics are gabapentanoids, serotonin–noradrenalin reuptake inhibitors (SNRIs), tricyclic antidepressants (TCAs), and topically administered medications [9]. Analgesics, such as NSAIDS, have no proven efficacy in CNP relief [11].

Second-line recommendations are combination/add-on therapy with first-line medications or tramadol and tapentadol [9].

Third-line recommendations are serotonin-specific reuptake inhibitors (SSRIs), anticonvulsants, and NMDA antagonists [9]. 

Fifth-line recommendations are low-dose opioids [9]. 

Fourth- and sixth-line recommendations are neurostimulation and targeted drug delivery, respectively [9], and are not covered in this review.

Studies investigating tailored treatments and standalone algorithmically tiered treatments in peripheral neuropathic pain conditions, such as herpes-, diabetes-, trigeminal, and HIV-associated neuropathic pain, have been conducted to some extent, but standalone studies of the treatment of CNP differentiated by underlying disorders and conditions are limited [3,5,12,13].

This review aims to compare current algorithmic analgesic drugs for the treatment of neuropathic pain in patients with an underlying CNS condition and to match current and other drug types with tier-recommended medications for pain relief and adverse effects.

## 2. Materials and Methods

### 2.1. Study Selection

This systematic review followed the “Reporting Checklist for Systematic Review” based on the PRISMA 2020 (Preferred Reporting Items for Systematic Reviews) guidelines [14] (Figure 1), and the data were submitted to Zenodo [15]. It was not registered in Prospero.

Relevant studies were identified through PubMed and EMBASE. The final search in both databases was performed on 19 November 2024. MeSH terms, EmTree terms, and text words were used in both search strategies (Figure 2).

The selected studies were transferred to EndNote 21 [16], where duplicates were removed. All titles/abstracts were screened for relevance by the primary reviewer (B.K.H.). In the case of conflict during full-text screening, the second reviewer (H.B.J.) made the final decision to include or omit the study.

The included studies fulfilled the inclusion criteria: (1) RCTs with (2) human patients, (3) aged 18 years or older, (4) written in the English language, and (5) with CNS-derived CNP.

Studies were excluded if they investigated neuropathic pain because of (1) malignancy, (2) iatrogenic origin, (3) discus prolapse, (4) peripheral origin, (5) non-pharmacological interventions, or (6) the study design did not center on treating CNP.

Statistical significance was defined as *p* < 0.05 (difference in pain or pain reduction). No study was excluded due to bias. One study was retracted by the publisher for plagiarism after publishing and was excluded from this review.

The risk of bias in the included studies was assessed by the primary reviewer (B.K.H.) using the 14-point US National Institutes of Health (NIH) Quality of Controlled Intervention Studies [17]. The studies were graded as good, fair, or poor.

### 2.2. Data Extraction and Quality Assessment

The data included first authors’ surname, year of publication, underlying condition(s), analgesic intervention, population after randomization, pain-measuring instrument(s), significance, effect size, and risk of bias.

### 2.3. Pain-Measuring Instruments

The included studies used a variety of instruments, such as questionnaires, surveys, and scales, to categorize and score different pains and changes.

Visual Analogue Scale (VAS)

A visual line spanning 0–10 or 0–100 mm, with 0 being the absence of pain and 10/100 mm being the worst pain imaginable [18].

McGill Pain Questionnaire (MPQ)

A 20-section questionnaire, where the patient marks which of the 78 descriptors describes the pain and PPI (present pain intensity) scored numerically from 0 (mild pain) to 5 (excruciating pain) [19].

Short-form McGill Pain Questionnaire (SFMPQ)

A 15-descriptor questionnaire on differing types of pain rated from 0 (none) to 3 (severe). Includes VAS and PPI [20].

Short-form McGill Pain Questionnaire 2 (SFMPQ2)

A 22-item descriptor and PPI scored numerically from 0 (no pain) to 10 (the worst pain imaginable) [21].

Borg Category Ratio Scale (CR-10)

A numerical scale ranging from 0 (no sensation/pain) to 10 (extreme sensation/pain), and an off-scale “•” for maximum sensation/pain [22].

Duration-Adjusted Average Pain Change (DAAC)

DAAC is a weighted average, proportional to participation duration, of observed and unobserved (missing) pain scores, numerically scored from 0 (no pain) to 11 (the worst pain imaginable) [23].

Numerical (Pain) Rating Scale (NRS/NPRS)

A numerically segmented scale in the range of 0–10 (or 0–100), with 0 being no pain and 10(100) being the worst pain imaginable [24]. It can be supplemented with faces showing increasing discomfort, thus being known as numerical pain rating scale-faces pain scale (NPRS-FPS).

Neuropathic Pain Scale (NPS)

A 10-descriptor scale differentiating types of pain and scoring them from 0 (no pain) to 10 (the worst pain imaginable) [25].

Neuropathic Pain Symptom Inventory (NPSI)

A questionnaire containing 10 items scored numerically from 0 (no pain) to 10 (the worst imaginable pain). The total pain intensity score is calculated by the sum of the 10 descriptors, plus 2 additional ones [26].

Pain Disability Index (PDI)

An index measuring the degree to which daily life is disrupted by chronic pain, consisting of 7 categories, scored numerically from 0 (no disability) to 10 (total disability) [27].

Multidimensional Pain Inventory (MPI)

A three-section questionnaire consisting of 52 items scored numerically from 0 (no impact/pain) to 6 (extreme impact/pain) [28].

Brief Pain Inventory (BPI)

A multiple descriptor questionnaire assessing the patient’s pain severity and the pain’s impact on everyday life, with most descriptors being scored from 0 (no impact/pain) to 10 (extreme impact/the worst pain imaginable) [29].

Short-Form Health Questionnaire (SF-36)

A 36-item questionnaire covering physical function, bodily pain, limitations due to health, limitations due to emotional problems, social functioning, fatigue, and own health perception. Bodily pain and pain interference with normal work are scored from 0 (none/not at all) to 5 (very severe/extreme) [30].

Complex Regional Pain Syndrome (CRPS) Score/Shoulder–Hand Syndrome Score

Three-subcategory questionnaire (Sensory: pain, hyperalgesia; Autonomic: distal edema; Motoric: painless passive range of motion) scored from 0 (no/none) up to 5 (most/massive). A total of 0–18 points is possible [31]. The pain subcategory was used in this review.

## 3. Results

A total of 851 studies were identified after removing duplicates. A total of 38 studies fulfilled the inclusion criteria and none of the exclusion criteria (Table 1). Please refer to Figure 1 for the screening process.

### 3.1. Investigated First-Line Treatment


**Tricyclic antidepressants (TCAs)**


Agarwal et al. (2017) [3] investigated amitriptyline in SCI patents and found a significant reduction (*p* = 0.000) in SFMPQ2-reported pain compared to placebo after three weeks of treatment. The study also investigated amitriptyline against lamotrigine and found no significant difference in pain relief (*p* > 0.005) between the two drugs.

Rintala et al. (2007) [42] investigated amitriptyline against gabapentin and placebo in SCI patents and found a significant reduction in VAS pain for certain subgroups after 8 weeks of treatment compared to the placebo (*p* = 0.035). Gabapentin was no more effective than the placebo for all patients (*p* = 0.97) as well as amitriptyline not being more effective than gabapentin (*p* = 0.061).


**Serotonin norepinephrine reuptake inhibitors (SNRIs)**


Iwaki et al. (2020) [57] investigated duloxetine in patients with Parkinson’s disease and found no statistically significant reduction in VAS pain.

Mahesh et al. (2023) [5] investigated duloxetine in patients with CPSP and found a significant reduction in self-reported pain in SFMPQ2 (*p* = 0.002) and NRS (*p* = 0.032) after 4 weeks of treatment compared to the placebo.

Vollmer et al. (2014) [53] investigated duloxetine in MS patients and found a significant reduction in NRS “average pain intensity” after 6 weeks of treatment compared to the placebo (*p* = 0.001).

Vranken et al. (2011) [6] investigated duloxetine in stroke and SCI patients and found no statistically significant reduction in VAS- (*p* = 0.056) or PDI (*p* = 0.063)-reported pain, but a significant reduction in SF-36 “bodily pain” (*p* = 0.035).


**Gabapentanoids**


Kaydok et al. (2014) [38] investigated pregabalin against gabapentin in SCI patients and found a significant difference reduction in VAS pain in the pregabalin group compared to the gabapentin group (*p* = 0.045), but no difference between the two groups after 8 weeks of treatment (*p* > 0.05).

Siddall et al. (2006) [44] investigated pregabalin in SCI patients and found a significant reduction in VAS (*p* < 0.001), NRS (*p* < 0.001) and SFMPQ (*p* ≤ 0.001–0.002) pain after 12 weeks compared to the placebo.

Cardenas et al. (2013) [23] investigated pregabalin SCI patients and found a significant reduction in DAAC pain (*p* = 0.003) and VAS pain (*p* = 0.007) after 16 weeks compared to the placebo.

Vranken et al. (2008) [8] investigated pregabalin in patients with mixed underlying conditions and found significant reduction in VAS-reported pain (*p* = 0.016) and in “bodily pain” of SF36 (*p* = 0.009) in compared to the placebo after 4 weeks.

Yilmaz et al. (2015) [47] investigated pregabalin against gabapentin in SCI patients and found no significant difference between the two drugs in VAS-reported pain (*p* = 0.06) or PDI-reported pain (*p* = 0.49) after 18 weeks of treatment.

Levendoglu et al. (2004) [40] investigated gabapentin in SCI patients and found a significant reduction in SCI patients in VAS-reported pain (*p* = 0.001) and pain intensity (*p* = 0.000), sharp pain (*p* = 0.000), hot pain (*p* = 0.001), deep pain (*p* = 0.000), and surface pain (*p* = 0.001) on NPS compared to the placebo after 8 weeks of treatment.

Tai et al. (2002) [45] investigated gabapentin in SCI patients and found no significant reduction on NPS against the placebo after 4 weeks of treatment (*p* > 0.05), except in the descriptor “unpleasant feeling” (*p* = 0.028).

As mentioned, Rintala at (2007) [42] investigated amitriptyline against gabapentin. Please refer to the *Tricyclic Antidepressants (TCAs)* Section for further information.


**Topicals**


Olusanya et al. (2023) [7] investigated locally applied capsaicin against a placebo in SCI patients and found a significant reduction in VAS and reported pain at 2 weeks (*p* = 0.005) and 4 weeks (*p* = 0.10) and a significant reduction in MPI “pain severity” at 2 and 4 weeks (*p* < 0.05).

### 3.2. Investigated Second-Line Treatment


**Tramadol**


Norrbrink et al. (2009) [41] investigated tramadol in patients with SCIs and found significant reduction in present pain intensity, general pain intensity, and the worst pain intensity on CR-10 (*p* < 0.05) and MPI scale pain severity after 4 weeks of treatment compared to the placebo (*p* < 0.05).

### 3.3. Investigated Third-Line Treatment


**Selective serotonin reuptake inhibitors (SSRIs)**


Foley et al. (2022) [49] investigated fluoxetine in MS patients and found no statistically significant improvement in pain on BPI pain interference (*p* = 0.57) or NPS (*p* < 0.14) after 96 weeks of treatment compared to a placebo, riluzole, or amiloride.


**Anticonvulsants**


Agarwal et al. (2017) [3] investigated lamotrigine in SCI patents and found a significant reduction (*p* = 0.000) in SFMPQ2-reported pain compared to the placebo after three weeks of treatment. The study also investigated amitriptyline against lamotrigine and found no significant difference in pain relief (*p* > 0.005) between the two drugs.

Falah et al. (2012) [48] investigated levetiracetam against a placebo in MS patients and found patients without touch-evoked pain had significantly better total pain (*p* = 0.0254) and pain relief (*p* = 0.0289) than the placebo on a 6-point verbal scale, but the total sample of patients had no statistically significant change in pain relief (*p* = 0.169), total pain intensity (*p* = 0.147), or any other outcome measured on 6-point verbal scale or 11-point NRS (*p* = 0.086–0.715) after 6 weeks of treatment.

Finnerup et al. (2009) [36] investigated levetiracetam against a placebo in MS patients and found no significant reduction in NRS pain (*p* = 0.46) or significant change measured on NPSI (*p* = 0.55–0.95) in SCI patients after 5 weeks of treatment.

Jungehulsing et al. (2013) [54] investigated levetiracetam against a placebo and found no significant improvement in NRS pain in patients with CPSP after 20 weeks of trial (*p* > 0.05).

Rossi et al. (2009) [50] investigated levetiracetam against a placebo in MS patients and found a significant reduction in VAS-reported mean pain intensity score, mean pain difference, and in the percentage of patients with a clinically significant pain compared to the placebo during the three-month trial (*p* < 0.05).

Kumru et al. (2018) [39] investigated baclofen against a placebo in SCI patients and found a significant reduction in NRS-reported pain after 1, 2, 4, and 8 h (*p* < 0.05), as well as in max. neuropathic pain at 4, 8, and 24 h; min. neuropathic pain at 4 and 24 h; average neuropathic pain at 4, 8, and 24 h; and current neuropathic pain at 4 and 8 h compared to the placebo on the BPI scale (*p* < 0.05) and a significant reduction in continuous pain at 4, 8, and 24 h (*p* < 0.05), significant reduction in paroxysmal pain at 4 and 8 h (*p* < 0.05), and a significant reduction in allodynia at 4 and 8 h (*p* < 0.05).


**N-Methyl-D-aspartate (NMDA) antagonists**


Kvarnström et al. (2004) [12] investigated ketamine against lidocaine and a placebo and found a significant reduction in VAS-reported pain the ketamine group against the placebo (*p* = 0.01), but not in the lidocaine group vs. the placebo (*p* = 0.60) during a 150 min trial with 40 min continuous infusion at the beginning.

Vranken et al. (2005) [1] investigated ketamine as an add-on therapy to existing analgesics in patients with mixed underlying conditions and did not find a significant reduction in VAS pain between groups (*p* > 0.05) but found a significant reduction in SF-36 “bodily pain” (*p* = 0.025) and PDI (*p* = 0.025) after one week of treatment.

Amr (2010) [32] investigated ketamine as add-on therapy to gabapentin in SCI patients and found a significant reduction in VAS pain compared to the placebo during 7 days of infusion and 2 weeks after the treatment was halted (*p* = 0.0002), but not after 3 and 4 weeks of halting the treatment (*p* = 0.25–0.54).

### 3.4. Investigated None-Tier Medication


**Cannabinoids**


Langford et al. (2013) [2] investigated THC/CBD in MS patients in combination with existing medication and found a significant reduction in NRS pain at 10 weeks (*p* = 0.028), but not at the primary endpoint of 14 weeks (*p* = 0.234).

Hansen et al. (2023) [58] compared THC, CBD, and THC+CBD in grouped MS and SCI patients and found no significant differences in NPSI pain between groups after 7 weeks of treatment (*p* > 0.05).

van Amerongen et al. (2018) [52] investigated THC against a placebo in MS patients and found a significant reduction in pain directly after the clinical administration of THC, but not when compared to pain diary for the full 4 weeks of treatment (*p* = 0.6581).

Rintala et al. (2010) [43] investigated dronabinol against a placebo in MS patients and found no significant difference in BPI-reported “average pain” after 17 weeks of treatment (*p* = 0.102).

Svendsen et al. (2004) [4] investigated dronabinol in MS patients and found a significant reduction in NRS-reported median spontaneous pain from baseline (*p* = 0.02), pain relief (*p* = 0.035), radiating pain (*p* = 0.039), pressure pain threshold (*p* = 0.036), and bodily pain on SF36 (*p* = 0.037) compared to a placebo after three weeks of treatment.

Turcotte et al. (2015) [51] investigated nabilone as an add-on treatment to gabapentin in MS patients and found a significant reduction in VAS-reported pain (*p* = 0.01) and VAS pain impact on daily activities (*p* = 0.01) compared to a placebo after adjusting for patient covariates after 9 weeks of treatment.


**Anti-inflammatory drugs**


Kalita et al. (2006) [56] investigated prednisolone against piroxicam in PS-CRPS patients and found a significant reduction in pain in in the prednisolone group after 1 month of treatment on the CRPS pain score (*p* < 0.0001).

Kalita et al. (2016) [13] investigated prednisolone against a placebo in PS-CRPS patients and found a significant reduction in baseline pain after 1 month of treatment on the CRPS pain score (*p* < 0.0001) and VAS pain in the prednisolone group (*p* < 0.001) compared to the baseline, and a significant reduction in CRPS pain after 1 month of treatment in the prednisolone group against the placebo (*p* < 0.01), but not in VAS (*p* > 0.05).

Ralph et al. (2020) [55] investigated etanercept against a placebo in patients with CPSP and found a significant reduction in VAS worst pain (*p* = 0.01) and average pain (*p* = 0.029) from baseline and in NPRS-FPS worst pain (*p* = 0.012) after 14 days of treatment.


**Sodium channel blockers**


Finnerup et al. (2005) [35] investigated lidocaine against a placebo in SCI patients and found a significant reduction in VAS-reported spontaneous pain for all patients (*p* = 0.01) and for subgroups with evoked pain (*p* < 0.05) and without evoked pain (*p* < 0.05) 25 min after IV lidocaine injection compared to the placebo.

As mentioned, Kvarnström et at (2004) [12] investigated ketamine against lidocaine and placebo. Please refer to the *N-Methyl-D-aspartate (NMDA) antagonists* Section for further information.

## 4. Discussion

The aim of this systematic review was to investigate analgesic medications in recognized guidelines and beyond in treating CNP. The algorithm recommended by Bates et al. (2019) [9] based on guidelines from national and international societies is generally confirmed by the current systematic review.

This discussion highlights significant findings in studies, drugs, and underlying conditions not stated in Bates et al. (2019) [9] or already investigated in Finnerup et al. (2015) [10] as well as differing pain relief in tier-different medication.

One study [42] investigated amitriptyline against gabapentin, both first-line recommendations, and a placebo in SCI patients, and found that, when the group was stratified for depressive symptomatology, gabapentin and amitriptyline were as effective as the placebo in patients with low depression, and amitriptyline and gabapentin showed greater pain relief in patients with high depression compared to the placebo, highlighting the need for further stratification into the severity of depression in studies to investigate the best first-line treatment in different groups.

One study [3] investigated amitriptyline, a first-line recommendation against lamotrigine, a third-line drug, in SCI patients, finding both drugs effective in reducing baseline pain, and no difference in pain reduction between the two after three weeks. The lamotrigine group reported no adverse effects in opposition to the amitriptyline group, where some adverse effects were reported. Studies with larger populations and longer durations are needed to corroborate these findings and to investigate if lamotrigine could be recommended before amitriptyline in SCI patients.

A study [47] on SCI patients compared gabapentin and pregabalin, both first-line recommendations, and found that both significantly reduced pain from baseline and found no difference in pain relief between the groups. Future studies should compare the adverse effect profile between the two in SCI patients and other CNP conditions.

Cannabinoids are not mentioned as recommended analgesic treatment in Bates et al. (2019) [9], and Finnerup et al. (2015) found weak recommendations against their use, citing high doses being needed for pain relief, the risk for dependency, and long-term adverse effects in adolescents and susceptible individuals [10]. Two studies found significant pain relief in MS patients using cannabinoids as standalone [4] and as add-ons to the recommended first-line treatment [51]; however, significant pain relief was not achieved in studies where SCI patients were stratified or grouped with MS patients [43,58].

Levetiracetam is also not mentioned in Bates et al. [9] and is strongly recommended against in Finnerup et al. (2015) [10] because of its generally negative results and safety concerns. One study [50] published in 2009 was not included in Finnerup et al. (2015) and found significant pain relief compared to a placebo in MS patients non-responsive to the now-recommended first- and second-line treatments, and no significant differences in adverse effects between groups. Future studies should investigate whether levetiracetam has potential in MS patients.

Duloxetine, a first-line recommendation, was found not to be significant in reducing VAS pain compared to a placebo in SCI and stroke patients in one study [6], but was found to significantly reduce NRS and SFMPQ2 pain in CPSP patients in another study [5], further suggesting that underlying conditions may respond differently to current treatments.

Parkinson’s disease is not mentioned in Bates et al. [9] or Finnerup et al. (2015) [10], and one study [57] found duloxetine was no better than the placebo in providing pain relief.

Prednisolone was found in two studies [13,56] to provide statistically significant pain relief in PS-CRPS patients, and etanercept was also found in a study [55] to significantly reduce pain in CPSP patients. Post-stroke pain conditions are, however, complex, possibly also involving trauma and peripheral nerve components [56], making it difficult to categorize these drugs as only improving CNP.

Two studies, one [35] investigating lidocaine injections and another [39] baclofen bolus injections in SCI patients, found significant reduction in pain; however, both studies only investigated pain relief in less than 48 h, making the analgesics not viable for daily pain relief treatment.

Three studies [34,46,49] investigated fluoxetine, riluzole, amiloride, NFX88, and lithium, drugs not included in the algorithm by Bates et al. [9] or in Finnerup et al. (2015) [10] and found no significant reductions in pain.

Some studies [6,13] found a significant outcome in one pain instrument but not in others, showing the need to conduct multitool studies when investigating CNP relief.

### Strengths and Limitations

The strengths of this review were the inclusion of all studies fulfilling the inclusion criteria up until 19 November 2024, which increased the number of identified articles. Studies were also rated for bias. This review included RCTs only, meaning that the included studies were more homogeneous and increasing comparability among studies based on study design.

This review was also broad enough to include underlying conditions and analgesics not formally recommended by the treatment algorithm.

Studies that grouped CNP and peripheral neuropathic pain patients were excluded, as well as studies with a grouping of pharmacological and nonpharmacological interventions together, as well as studies investigating malignant or iatrogenic CNP, limiting the number of included studies.

The number of participants in the included studies varied considerably, making direct comparisons among outcomes difficult.

The differing pain scales and questionnaires in the included studies made direct comparisons among studies difficult, but we attempted to address this by showing numerical changes and percentage changes in pain on the different scales, where applicable.

Most of the included studies investigated SCI patients, limiting outcomes for other patient groups.

## 5. Conclusions

This systematic review agrees with the algorithmic guidelines for CNP management set by Bates et al., but finds that certain underlying conditions may benefit from specified approaches compared to the general treatment algorithm, as certain conditions appear to respond differently and exhibit differing adverse effects. More studies are needed to clarify this aspect.

## Figures and Tables

**Figure 1 neurolint-17-00077-f001:**
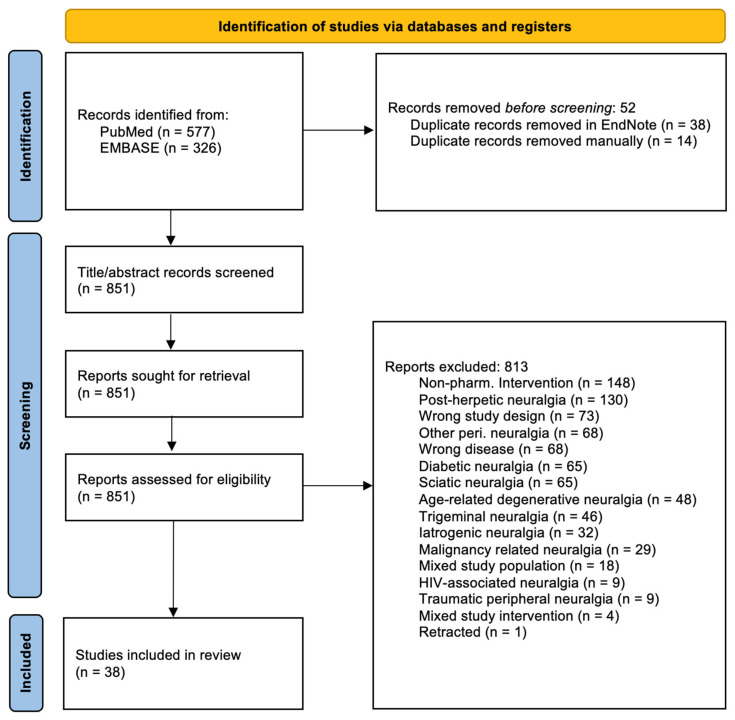
PRISMA 2020 flow diagram showing the included and excluded studies and the reasons for exclusion.

**Figure 2 neurolint-17-00077-f002:**
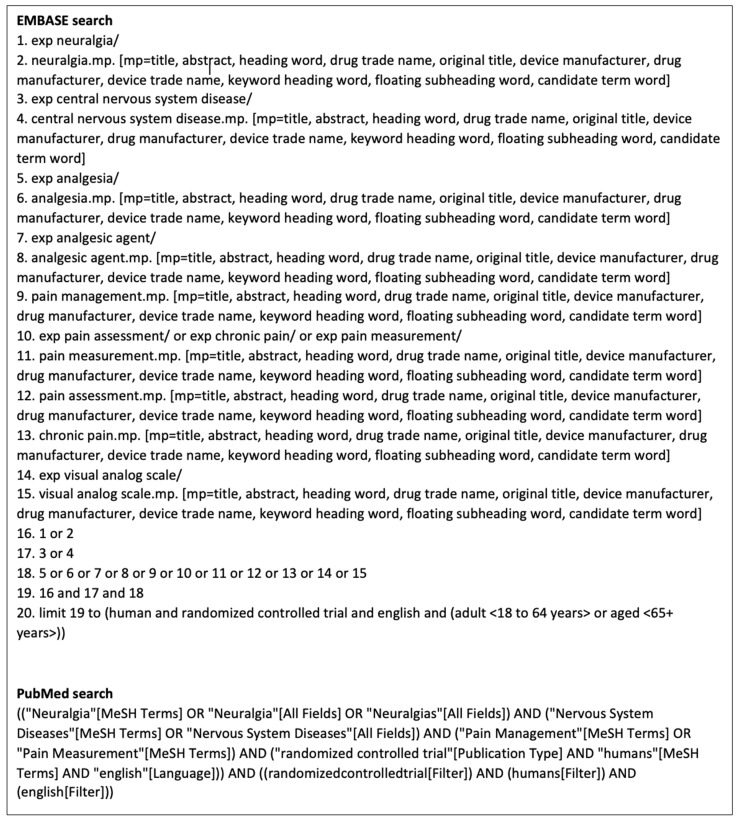
Literature search string from EMBASE and PubMed.

**Table 1 neurolint-17-00077-t001:** Overview of the included studies.

Name, Year	Underlying Condition(s)	Analgesic(s)	Pop.	Pain-Measuring Instrument(s)	Significance	Effect Size	Risk of Bias
Agarwal et al., 2017 [3]	SCI	Amitriptyline Lamotrigine	147	SFMPQ2, 0–10	Pain reduction from baseline, *p* = 0.000 * Difference between drugs, *p* = 0.648–0.819	Amitriptyline: Days 7–14–21 Mean: 0.2149, s.d. 0.2075 Mean: 0.3593, s.d. 0.1939 Mean: 0.4312, s.d. 0.1812 Lamotrigine Days 7–14–21 Mean: 0.2048, s.d. 0.2245 Mean: 0.3622, s.d. 0.2403 Mean: 0.4432, s.d. 0.1812	Fair
Amr, 2010 [32]	SCI	Ketamine add-on to gabapentin	40	VAS, 0–10	Reduced pain in both groups, *p* < 0.05 * Reduction in pain vs pre-treatment, *p* = 0.0001 * Reduction in pain, 7 days of treatment + 2 weeks after stopping treatment, *p* = 0.0002 * 3 + 4 weeks after stopping treatment, *p* = 0.25–0.54	VAS and % difference Days 1–7: 27.3 ± 8.9, 72.3 ± 5.4 (90.36%) 14.0 ± 5.6, 42.5 ± 14.5 (100.89%) 1st week, 2nd week after last infusion 21.50 ± 6.39, 43.00 ± 4.05 (66.67%) 22.4 ± 7.54, 44.00 ± 4.61 (65.06%)	Good
Cardenas et al., 2013 [23]	SCI	Pregabalin	220	DAAC, 0–10 NRS, 0–10	DAAC, *p* = 0.003 * NRS pain, change from baseline, *p* = 0.007 *	DAAC, least squares, differences between groups: Placebo −1.07 Pregabalin −1.66 % difference (43.22%) Pain, NRS, least squares, difference in change from baseline pain: Placebo: −1.22 Pregabalin: −1.92 % difference (44.59%)	Fair
Chun et al., 2019 [33]	SCI	Botox-A	8	NPRS, 0–10	Underpowered	N/A	Fair
Escribá et al. 2024 [34]	SCI	NFX88 add-on to pregabalin	61	VAS, 0–10	Total patients, *p* < 0.2	VAS, mean (nom. % chance) Visit 1, Visit 4 Placebo: 7.17, 5.50 (−23.29%) 1.05 g/day: 6.62, 5.92 (−10.57%) 2.1 g/day: 6.63, 4.32 (−34.84%) 4.2/day: 7.02, 6.01 (−14.39%)	Good
Finnerup et al., 2005 [35]	SCI	Lidocaine	26	VAS, 0–10	Total patients, *p* = 0.01 *	VAS average, 0–25 min: All patients 57.08, 39.00 % difference (−37.64%)	Good
Finnerup et al., 2009 [36]	SCI	Levetiracetam	36	NRS, 0–10 NPSI, 0–10	NRS, *p* = 0.46 *NPSI*, *p* = 0.55–0.95	Median NRS pain, placebo % change (−16.67%) Median NRS pain, levetiracetam: % change (0.00%) Levetiracetam vs. baseline, combined NPSI % change, (−2.11%)	Good
Han et al., 2016 [37]	SCI	Botox-A	40	VAS, 0–100 mm SFMPQ, 0–3	VAS, *p* = 0.0027–0.0053 * SFMPQ Sensory, *p* = 0.0033 * Affective, *p* = 0.0086–0.0131 * Total scores, *p* = 0.0008–0.0197 *	VAS, Botox 0–4–8 weeks, 85.1 ± 13.6 (−21.86%) 66.5 ± 20.7 (−4.06%) 63.8 ± 27.5 VAS, Placebo, 0–4–8 weeks, 7.1 ± 14.0 (−3.37%) 74.5 ± 16.0 (3.09%) 76.8 ± 20.4	Good
Kaydok et al., 2014 [38]	SCI	Gabapentin Pregabalin	28	VAS, 0–10 NPS, 0–10	VAS, both drugs compared from baseline to 8 weeks, *p* < 0.001 * Pregabalin, pain reduction compared to gabapentin at 4 weeks, *p* = 0.045 * Pregabalin, pain reduction compared to gabapentin at 8 weeks, *p* > 0.05 NPS pregabalin compared to gabapentin, *p* > 0.05	VAS, 0–4–8 week: Gabapentin: 7.78 ± 1.27, 4.36 ± 1.30 (−43.96%), 3.57 ± 1.21 (−54.11%) Pregabalin: 8.05 ± 1.26, 3.68 ± 1.15 (−54.29%), 3.36 ± 1.11 (−58.26%) NPS “Pain intensity” 0–8 week: Gabapentin: 52.1% ± 18.1 Pregabalin: 54.5% ± 17.8	Poor
Kumru et al., 2018 [39]	SCI	Baclofen	13	NRS, 0–10 NPSI, 0–10 BPI, 0–10	NRS, pain: 1, 2, 4, and 8 h from baseline, *p* < 0.05 * NPSI, continues pain: 4, 8, and 24 h from baseline, *p* < 0.05 * NPSI, paroxysmal pain: 4 and 8 h from baseline, *p* < 0.05 * NPSI, allodynia: 4 and 8 h from baseline, *p* < 0.05 * BPI, max. neuropathic pain: 4, 8, and 24 h, *p* < 0.05 * BPI, min. neuropathic pain: 4 and 24 h, *p* < 0.05 *	NRS and nom. % change from baseline 4 h, 8 h, and 24 h: Baclofen: 3.0 ± 2.2 (−51%), 2.6 ± 2.2 (−59%), 4.5 ± 1.9 (−23%) NPSI, continuous pain and nom., 2.0 ± 1.7, 1.9 ± 1.5, 3.0 ± 1.5 NPSI, paroxysmal pain and nom., 0.4 ± 1.0, 0.3 ± 0.6, 1.8 ± 2.6 NPSI, allodynia and nom. % change from baseline, 1.0 ± 1.2, 0.6 ± 1.0, 1.7 ± 1.1 BPI, baclofen % of improvement with treatment, 68.8% ± 23.0, 62.5% ± 30.6, 50.0% ± 25.6	Fair
Kvarnström et al., 2004 [12]	SCI	Ketamine Lidocaine	10	VAS, 0–100 mm	Ketamine vs. placebo, *p* < 0.01 * Lidocaine vs. placebo, *p* < 0.60	Mean VAS for all patients before infusion: 4.8 Mean % max pain reduction, ketamine, (−38%) Mean % max. pain reduction, lidocaine, (−10%) Mean % max. pain reduction, placebo, (−3%)	Good
Levendoglu et al., 2004 [40]	SCI	Gabapentin	20	VAS, 0–100 NPS, 0–10	*p* = 0.000–0.001 *	Mean VAS points reduction during 0–8 week, gabapentin group: (60.7 ± 12.7%) Mean VAS points reduction during 0–8 week, placebo group: (10.3 ± 2.8%) Pain intensity, gabapentin group, 0–4–8 week, % change 0–8 week: 8.5 ± 0.9, 4.8 ± 1.1, 3.2 ± 1.2 (61.9 ± 14.3%) Pain intensity, placebo group, 0–4–8 week, % change 0–8 week: 6.0 ± 3.9, 5.3 ± 3.5, 5.5 ± 3.6 (13.2 ± 4.8%)	Good
Norrbrink et al., 2009 [41]	SCI	Tramadol	36	MPI, 0–6 CR-10, 0–10	MPI, *p* < 0.05 * CR-10, *p* < 0.05 *	Tramadol median, 0–4 week, Present pain: 3, 3 General pain: 4, 3 The Worst pain: 7, 5 MPI-Pain: 3, 2.5 Placebo median, 0–4 week, Present pain: 5, 5.5 General pain: 7, 6.5 The Worst pain: 9, 8 MPI-Pain: 4, 4.13	Poor
Olusanya et al., 2023 [7]	SCI	Capsaicin	34	MPI, 0–6 VAS, 0–10	MPI, 2 weeks, *p* < 0.05 * MPI, 4 weeks, *p* < 0.05 * VAS, 2 weeks, *p* = 0.029 * VAS, 4 weeks, *p* = 0.049 *	VAS % change, 2nd week: −35% 4th week: −29%	Fair
Rintala et al., 2007 [42]	SCI	Amitriptyline Gabapentin	38	VAS, 0–100 mm NRS, 0–10	VAS Amitriptyline vs. placebo from baseline, *p* = 0.035 * Gabapentin vs. placebo, *p* = 0.97 Amitriptyline vs. gabapentin, *p* = 0.061	Mean decrease in average VAS pain, % change in VAS average pain, 0–8 week: Group 1, minor depression: Amitriptyline −1.58 (−31.5%) Gabapentin −0.84 (−134.9%) Placebo −0.40 (−16.5%) Group 2, major depression: Amitriptyline −3.21 (−40.6%) Gabapentin −0.70 (−11.3%) Placebo −0.74 (−8.7%)	Fair
Rintala et al., 2010 [43]	SCI	Dronabinol	7	BPI, 0–10	*p* = 0.102	Pain intensity at its worst, 8.1 ± 1.6 Mean change from baseline, dronabinol group, 0.20 ± 0.837 Mean change from baseline, placebo group, −1.80 ± 2.49	Fair
Siddall et al., 2006 [44]	SCI	Pregabalin	137	NRS, 0–10 VAS, 0–100 SFMPQ, 0–3	VAS, *p* < 0.001 * NRS, *p* < 0.001 * *SFMPQ*, *p* ≤ 0.001–0.002 *	Placebo 0–12 week, NRS: 6.73, 6.27 Sensory: 14.0, 13.3 VAS: 73.1, 68.5 Pregabalin 0–12 week NRS: 6.54, 4.62 Sensory: 13.4, 9.3 VAS: 69.1, 49.2	Fair
Tai et al., 2002 [45]	SCI	Gabapentin	14	NPS, 0–10	4th week for “unpleasant feeling” *p* = 0.028 * Rest of NPS, *p* > 0.05	Average unpleasantness (0 week), 7.26 Average unpleasantness (4th week), Gabapentin, 3.60; Placebo, 5.33	Fair
Yang et al., 2012 [46]	SCI	Lithium	40	VAS, 0–100	0–6 months: VAS, *p* = 0.237–0.97	VAS Pain, 0–6 weeks, Placebo, −1 ± 3.97 Lithium, 8.833 ± 14.861 VAS Pain, 6 weeks–6 months, Placebo, 0.778 ± 17.176 Lithium, 9.389 ± 15.232	Fair
Yilmaz et al., 2015 [47]	SCI	Gabapentin Pregabalin	21	VAS, 0–10	*p* > 0.05	Pain VAS, 1st period, 0–8 weeks Pregabalin 7.05 ± 1.92, 3.83 ± 3.40 Gabapentin 7.02 ± 1.63, 4.77 ± 2.77 Pain VAS, 2nd period, 0–8 weeks Pregabalin 7.42 ± 1.66, 2.53 ± 1.98 Gabapentin 8.60 ± 1.34, 5.00 ± 1.41	Poor
Falah et al., 2012 [48]	MS	Levetiracetam	37	6-point verbal scale	Total patients, *p* = 0.157–169	“Total pain” 0–6 week: Levetiracetam, 5.8, 5.3 % change (−8.62%) Placebo: 5.8, 5.7 % change (−1.72%) % difference (7.27%) “Pain relief” 0–6 week: Levetiracetam 2.4, 2.1 % change (12.5%)	Good
Foley et al., 2022 [49]	MS	Fluoxetine Riluzole Amiloride	445	NPS, 0–10 BPI, 0–10	NPS, *p* = 0.14 *p* = 0.26 *p* = 0.27 BPI, *p* = 0.57 *p* = 0.92 *p* = 0.18	NPS pain relief: Fluoxetine–placebo: 0.52 Riluzole–placebo: 0.40 Amiloride–placebo: 0.38 BPI Pain interference: Fluoxetine–placebo: 0.18 Riluzole–placebo: −0.03 Amiloride–placebo: 0.42	Fair
Langford et al., 2013 [2]	MS	Add-on THC/CBD	393	NRS, 0–10 SF-36, 0–5	NRS: *p* = 0.234 SF-36 “Bodily pain”, *p* = 0.494	Bodily pain, THC/CBD, Placebo, % difference 11.36, 10.01, 12.63%	Good
Rossi et al., 2009 [50]	MS	Levetiracetam	20	VAS, 0–100 mm	Mean pain intensity, *p* < 0.005 * Mean pain difference, *p* < 0.005 * % of patients with clinically significant pain reduction *p* < 0.05 *	Mean pain VAS score Placebo, Levetiracetam T1: −12.5%, −18.2% T2: −12.5%, −72.7% T3: −14.3%, −81.8%	Poor
Svendsen et al., 2004 [4]	MS	Dronabinol	24	NRS, 0–10	NRS: *p* = 0.02–0.039 * SF-36 Bodily pain, *p* = 0.037 *	Active–placebo, placebo–active Spontaneous pain, −1.0, 0, 0, −1.5 Pain relief, 3.0, 0.5, 0, 4.0 Bodily pain, 41.0, 26.5, 42.0, 61.0	Good
Turcotte et al., 2015 [51]	MS	Nabilone add-on to gabapentin	15	VAS, 0–100 mm	VAS pain, *p* < 0.01 * VAS pain impact, *p* < 0.01 *	N/A	Fair
van Amerongen et al., 2018 [52]	MS	THC	24	NRS, 0–10	*p* = 0.6581	NRS Pain, 0–4 week Placebo, 2.57 THC, 2.10 % difference (−20.13%)	Good
Vollmer et al. 2014 [53]	MS	Duloxetine	239	BPI, 0–10	*p* = 0.001 *	Average pain intensity, baseline: Duloxetine, 6.5; Placebo, 6.3 Mean change in weekly API: Duloxetine, −1.83; Placebo, −1.07 % difference (−41.53%)	Fair
Jungehulsing et al., 2013 [54]	CPSP	Levetiracetam	42	NRS, 0–10	*p* > 0.05	N/A	Fair
Mahesh et al., 2023 [5]	CPSP	Duloxetine	82	NRS, 0–10 SFMPQ2, 0–10	NRS, *p* = 0.002 * SFMPQ2, *p* = 0.032 *	NRS, 0–4 week and % change Duloxetine group: 6.51 ± 1.03, 3.02 ± 1.70, (−53.61%) Placebo group: 6.37 ± 1.41, 4.40 ± 1.77, (−30.93%) SFMPQ2, 0–4 week, % change, Duloxetine group: 19.53 ± 6.34, 8.85 ± 4.02, (−54.69%) Placebo group: 20.29 ± 7.10,13.29 ± 5.74, (−34.50%)	Good
Ralph et al., 2020 [55]	CPSP	Etanercept	20	VAS, 0–100 NPRS-FPS, 0–10	VAS: “Worst pain”, *p* = 0.01 * “Average pain”, *p* = 0.029 * NPRS-FPS: *p* = 0.012 *	NPRS-FPS “Worst pain” Etanercept group Median decrease: 27.5 points Mean decrease: 33.5 ± 10.4 Placebo group Median decrease 17.5 Mean decrease 13.33 ± 6.7	Good
Kalita et al., 2006 [56]	PS-CRPS	Prednisolone Piroxicam	60	CRPS, 0–5	Prednisolone pain reduction against piroxicam *p* < 0.0001 *	CRPS pain, prednisolone group, 0–1 month: 3.97 ± 0.85, 1.13 ± 1.31 (−71.54%) CRPS pain, piroxicam group, 0–1 month: 4.00 ± 0.87, 3.67 ± 1.35 (−8.25%)	Fair
Kalita et al., 2016 [13]	PS-CRPS	Prednisolone	77	VAS, 0–10 CRPS, 0–5	Prednisolone vs. baseline 0–1 month VAS, *p* < 0.001 * CRPS, *p* < 0.001 * Prednisolone vs. placebo, 0–1 month VAS, *p* > 0.05 CRPS, *p* < 0.01 *	VAS, 0–1 month, nom. % change Prednisolone continued group, 3.6 ± 1.1, 2.4 ± 1.0, (−33.33%) Prednisolone discontinued group, 3.5 ± 1.0, 4.9 ± 2.1, (−40.00%) CRPS “pain score” 0–1 month, nom. % change: Prednisolone-continued group, 4.3 ± 0.9, 0.8 ± 0.5, (−81.40%) Prednisolone-discontinued group, 4.6 ± 1.5, 2.1 ± 1.2, (−54.35%)	Poor
Iwaki et al., 2020 [57]	Parkinson’s disease	Duloxetine	47	VAS, 0–100 mm	*p* > 0.05	Mean change, % change from baseline Duloxetine, −0.83, (−2.28%) Placebo, −1.91, (−4.99%)	Fair
Hansen et al., 2023 [58]	SCI MS	THC CBD THC+CBD	134	NPSI, 0–10	*p* > 0.05	Difference from baseline, NPSI, total difference: THC, −1.3 CBD, −1.5 THC+CBD, −0.8	Fair
Vranken et al., 2011 [6]	SCI Stroke	Duloxetine	48	VAS, 0–10 SF-36, 0–5 PDI, 0–10	VAS: *p* = 0.056 SF-36, bodily pain: *p* = 0.035 * PDI: *p* = 0.063	VAS mean pain intensity, 0–8 weeks: Placebo, 7.2 ± 0.8, 6.1 ± 1.7 (−15%) Duloxetine, 7.1 ± 0.8, 5.0 ± 2.0 (−29.6%) SF-36, 0–8 weeks: Placebo, 31 ± 12, 35 ± 14 Duloxetine, 33 ± 13, 45 ± 17 PDI, 0–8 weeks: Placebo, 38 ± 14.3, 36 ± 13.3 Duloxetine, 33 ± 11.2, 28 ± 12.2	Good
Vranken et al., 2005 [1]	Stroke MS Parkinson’s disease Thalamus lesion Brainstem pathology SCI	Ketamine	33	VAS, 0–10 SF-36, 0–5 PDI, 0–10	VAS, *p* > 0.05 SF-36, bodily pain 75 mg/day after 7 days of treatment, *p* = 0.025 * PDI, 75 mg/day after 7 days of treatment, *p* = 0.025 *	VAS mean pain intensity, 0–1 week: Placebo, 7.1, 6.4, (−9.86%) Ketamine 50 mg, 7.3, 6.2, (−15.07%) Ketamine 75 mg, 7.3, 5.7, (−21.92%) SF-36, 0–1 week: Placebo, 16.5 ± 14.2, 22.4 ± 16.2 Ketamine 50 mg, 14.5 ± 13.4, 34.4 ± 11.7 Ketamine 75 mg, 13.1 ± 9.6, 66.6 ± 13.1 PDI, 0–1 week: Placebo, 43.6 ± 10.2, 42.7 ± 11.1 Ketamine 50 mg, 45.5 ± 11.9, 45.0 ± 13.2 Ketamine 75 mg, 45.6 ± 5.7, 19.5 ± 12.9	Good
Vranken et al., 2008 [8]	Stroke Thalamus lesion Brainstem pathology SCI	Pregabalin	40	VAS, 0–10 SF36, 0–5 PDI, 0–10	From baseline: VAS, *p* = 0.01–0.016 * SF-36, bodily pain, *p* = 0.009 * PDI, *p* = 0.111	VAS mean pain intensity, 0–4 weeks: Placebo, 7.4 ± 1.0, 7.3 ± 2.0, (−1.35%) Pregabalin, 7.6 ± 0.8, 5.1 ± 2.9, (−32.90%) SF-36, 0–4 weeks: Placebo, 26.2 ± 15.4, 27.8 ± 19.4 Pregabalin, 30.7 ± 16.1, 46.3 ± 20.2 PDI, 0–4 weeks: Placebo, 41.7 ± 14.8, 43.3 ± 14.7 Duloxetine, 39.9 ± 13.2, 35.7 ± 14.9	Good

* *p* < 0.05.

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
