# Peer review of "Medications for Managing Central Neuropathic Pain as a Result of Underlying Conditions—A Systematic Review"

_2035-8377, 2025, doi:10.3390/neurolint17050077_

Round 1
Reviewer 1 Report
Comments and Suggestions for Authors
Neurology International
Medications for Managing Central Neuropathic Pain – A Systematic Review
Bjarke Kaae Houlind and Henrik Boye Jensen
The authors of this article reviewed the current literature (up to November 2024) on analgesic treatments of central neuropathic pain associated with several clinical conditions and used certain algorithmic tools for analysis of the selected studies.
General Comment:
This systematic review is a confirmatory project. It agrees with the algorithmic guidelines for central neuropathic pain (CNP) management as set by previous studies 5 and 10 years ago (Refs. 9 and 10, respectively).
Comments:
Abstract – Include: (1) the names of the clinical conditions associated with (CNP); (2) the period in years of the selected publications.
Introduction – Page 2, Ln. 54: Define “algorithmic analgesic drugs”.
Table 1 – It would greatly help to arrange the list of references according to clinical condition-associated CNP rather than just alphabetically.
Author Response
Dear reviewer
Thank you very much for your comments, which we have taken to heart, and made adjustments accordingly. Please see below. Please excuse that certain pages have switched from portrait to landscape, and will be restored before final publication.
Comments 1:
Abstract – Include: (1) the names of the clinical conditions associated with (CNP);
(2) the period in years of the selected publications.
Response 1:
Clinical conditions added, and period of years for selected publications included.
Comments 2:
Introduction – Page 2, Ln. 54: Define “algorithmic analgesic drugs”.
Response 2: wording has been specified and changed.
Comments 3:
Table 1 – It would greatly help to arrange the list of references according to clinical condition-associated CNP rather than just alphabetically.
Response 3:
Table has been adjusted and arranged according to clinical condition rather than authors last name alphabetically.
Please let us know if these changes are acceptable to you.
Best regards,
Bjarke Houlind
Reviewer 2 Report
Comments and Suggestions for Authors I have read and reviewed this manuscript with great interest, and overall, from this reviewer's perspective, it is a systematic review that has been well-planned and executed. Overall, it is a study with refreshingly simple wording that is easy to understand. Other strengths of the manuscript that I can highlight are the following: the introduction provides sufficient background and includes pertinent references, the research design is adequate, and the methods are repeatable and correctly described. The results support the conclusions.
Author Response
Dear reviewer
Thank you so much for your kind words. We are very happy you enjoyed our work.
Best regards,
Bjarke Houlind
Reviewer 3 Report
Comments and Suggestions for Authors
The title recalls an article already published recently: https://doi.org/10.1155/2021/6656863. Unfortunately, your systematic review has not been registered on PROSPERO.
L10-12 are not a goal of a systematic review
L13 why not Scopus or WoS? The description of the population is missing (L77: all disorders?), above all as an intervention what is meant by medications, what controls have been hypothesized, and then above all as outcomes which have you thought of investigating? In short, the PICO is missing. If reported, use the RoB-2.
Results, unfortunately I was not able to use the strings in the methods, but I believe that L17: 903 are not RCTs, but records from the search strategy, L18: progressive and stable neurological diseases, is it not clear? which ones? are they relatable? here too, a clear PICO is missing.
L20 I recommend describing the conclusions of your study on the major findings of the systematic review, not conjectural comparisons to neuropathic pain management algorithms.
L81: Statistical significance was defined as P < 0.05. in which scenario?
L85 should be described after with the risk of bias used (RoB-2, I suggest)
L92 The outcomes are unstructured, the intervention and control for the pico are missing in the methods.
L162 compared to the abstract have become: 851 records
lines before L357 there is no risk of bias analysis...
L358: the beginning of the discussion by convention should be the paraphrase of the objective, among other things confirming national guidelines should not be the focus of study of your manuscript and systematic review
Author Response
Dear reviewer
Thank you very much for your comments, which we have taken to heart, and made adjustments accordingly. Please see below. Please excuse that certain pages have switched from portrait to landscape, and will be restored before final publication.
Comment 1: The title recalls an article already published recently: https://doi.org/10.1155/2021/6656863. Unfortunately, your systematic review has not been registered on PROSPERO.
Response 1: The title has been altered to differentiate from already published article.
Comment 2: L10-12 are not a goal of a systematic review.
Response 2: Goal more clearly stated and highlighted.
Comment 3: L13 why not Scopus or WoS?
Response 3: Thank you for your suggestion. For this systematic review Embase and Pubmed was deemed sufficient.
Comment 4: The description of the population is missing (L77: all disorders?), above all as an intervention what is meant by medications, what controls have been hypothesized, and then above all as outcomes which have you thought of investigating? In short, the PICO is missing. If reported, use the RoB-2.
Response 4: all patient exhibiting CNP derived from CNS were eligible, if fitting the inclusion criteria and none of the exclusion criteria.
Comment 5: Results, unfortunately I was not able to use the strings in the methods, but I believe that L17: 903 are not RCTs, but records from the search strategy
Response 5: 903 records were labeled as RCTs in Embase and Pubmed.
Comment 6: L18: progressive and stable neurological diseases, is it not clear? which ones? are they relatable? here too, a clear PICO is missing.
Response 6: by progressive neurological diseases, disease such as MS and Parkinsons disease, and by stable diseases/conditions are stroke and SCI. Their relatability in this case comes from that they all can cause central neuropathic pain.
Comment 7: L20 I recommend describing the conclusions of your study on the major findings of the systematic review, not conjectural comparisons to neuropathic pain management algorithms.
Response 7: Thank you for your recommendation. Conclusion has been added.
Comment 8: L81: Statistical significance was defined as P < 0.05. in which scenario?
Response 8: difference in pain or pain reduction has been added.
Comment 9: L85 should be described after with the risk of bias used (RoB-2, I suggest)
Response 9: RoB was moved to "study selection".
Comment 10: L92 The outcomes are unstructured, the intervention and control for the pico are missing in the methods.
Response 10: Pain measuring instruments varied widely between included studies, and we found we needed to include all of them. We attempted to highlight this in strength and limitations: "The differing pain scales and questionnaires in the included studies made direct comparison between studies difficult but was attempted mended by showing numerical change and percentage change in pain on the different scales where applicable."
Comment 11: L162 compared to the abstract have become: 851 records
lines before L357 there is no risk of bias analysis...
Response 11: 851 records were identified after 52 duplicates were removed.
Comment 12: L358: the beginning of the discussion by convention should be the paraphrase of the objective, among other things confirming national guidelines should not be the focus of study of your manuscript and systematic review
Response 12: paraphrasing of objective has been moved to the beginning of discussion.
Please let us know if these changes are acceptable to you.
Best regards,
Bjarke Houlind
Round 2
Reviewer 3 Report
Comments and Suggestions for Authors
Dear Authors, unfortunately the methodological gaps of your systematic review are insurmountable, from the failure to consider the PRISMA guidelines, the registration on PROSPERO for similarity with another manuscript.